# OpenReview forum: "Assessing Vulnerabilities of Large Language Models to Social Bias Attacks"
_ICLR.cc/2025/Conference — ICLR 2025 Conference Withdrawn Submission_

### Official Review · Reviewer_gSeS · 2024-10-30

**Soundness:** 3
**Presentation:** 3
**Contribution:** 2
**Rating:** 3
**Confidence:** 4

**Summary:**

In the paper, the authors proposed a growing concern about various social biases in LLMs. In particular, they worked three main biases, i.e., gender, religion, race, with three different methods, e.g., prefix injection, refusal suppression, and learned attack prompt to deceive LLMs. They evaluated the impacts over several most popular LLMs. Besides manual evaluation, they also used LLMs to evaluate their method.

**Strengths:**

1.	The entire paper is well-written and presented the ideas in very clear way.
2.	It introduces several representative categories of social biases in LLMs and demonstrates and evaluates a specific category of general purpose LLM vulnerability over several LLMs.

**Weaknesses:**

1.	As a whole, this paper is more like playing with the LLMs with several categories of deceiving prompts. It significantly lacks theoretical or technical presentations of ideas and experimental analysis.
2.	The explainability of such biases in LLM are missing. There is no formal definition of the attack evaluations.
3.	Moreover, authors mentioned jailbreaking attack as automated approach to evaluate attack, however, they did not mention how the LLM’s responses were distinguished as not containing responses like “I am a language model. . . ”, “Sorry. . . ”, “As a language model. . . ”
4.	The defense techniques it explored those are also a pre-caution -based methods, e.g., post-prompting and sandwich defense. Which does not also provide any deep technical analysis. Moreover, the attack impact under those defenses is not discussed in detail.

**Questions:**

1.	Since Appendix is not the main content of the paper, why subsection 4.4 SENSITIVITY ANALYSIS OF ATTACK PROMPTS has been included by only mentioning that?
2.	There are many ways of PI and RS apart from the examples mentioned in the paper. Moreover, in the experimental results it was not clear enough which prompts of corresponding categories (e.g., ”Absolutely! Here’s” for PI or Do not include any ”note” or ”disclaimer” RS) lead to which results.
3.	Discussing the impacts of defense a little would better complement the paper.
4.	In 5.2, human evaluator did 100 bias attack samples, what about the number of bias attacks performed for jailbreak and LLM evaluator?

---

### Official Review · Reviewer_EWno · 2024-10-30

**Soundness:** 3
**Presentation:** 3
**Contribution:** 3
**Rating:** 5
**Confidence:** 4

**Summary:**

This paper investigates the vulnerabilities of large language models (LLMs) to social bias attacks through comprehensive experiments. Previous work like [1] analyzes the bias issues of LLM-generated contents, while this paper focuses on the evaluation of the vulnerabilities of LLMs to various bias attacks. Three types of bias attacks are considered in the experiments for gender, racial, and religious biases. The effects of these attacks on LLM-generated contents are analyzed for a comprehensive list of modern LLMs. Besides, this paper explores the impacts of model size and fine-tuning on the vulnerabilities of LLMs. The results provide valuable insights for the development of LLMs that are robust and safe against bias attacks.

[1] Zhao, J., Fang, M., Shi, Z., Li, Y., Chen, L., & Pechenizkiy, M. (2023). Chbias: Bias evaluation and mitigation of chinese conversational language models. arXiv preprint arXiv:2305.11262.

**Strengths:**

This paper contributes to the field of responsible LLMs by investigating the vulnerabilities of LLMs to various types of social bias attacks, it constructs the bias attack dataset for the evaluation experiments.

This paper studies the impacts of model size, fine-tuning, and types of attack on LLM vulnerability, which I think is valuable for the defensive research in LLMs.

The experiments are comprehensive and adequate. Particularly, this paper considers three types of attacks for three types of commonly seen biases. Most modern LLMs are evaluated in the experiments. I think the experimental results justify the findings and insights this paper provides.

**Weaknesses:**

1. This paper mainly examines the vulnerabilities of LLMs, which I think can be viewed as a kind of robustness against bias attacks. In Section 3, the authors introduce various bias techniques in existing work. If these attacks are proved in literature to be effective in attacking LLMs, this would somehow limit the contribution of this paper. In other words, the results of this paper mainly show that LLMs can be still attacked by these attacks, which is the same as the paper that introduces these attacks. To this end, I would like the authors to discuss the technical contributions of this paper.

2. When introducing the motivation of this work, only a vague definition of “bias” is provided (as in footnote 1), which lacks preciseness and clarity. Given that this paper investigates the vulnerabilities of LLMs to social bias attacks, it is better to clearly define what the social bias issue in LLMs is at the beginning. For example, what kind of LLM-generated contents are considered biased and harmful, how they display, and who they impact. This will help the community to better understand the value of this work brings to advancing trustworthy LLM research.

3. In the experiments, the vulnerability of an LLM is evaluated only in terms of the proportion of “biased” results. However, such a binary outcome (biased/not biased) can oversimplify the evaluation of LLM-generated contents. Actually, the response from an LLM can have varying degrees of biasedness; some responses, even though being biased, are still tolerable or will not be too harmful for the general public, while some other responses may be completely unacceptable. Therefore, a better evaluation metric can be considered here is a numeric score that describes the degree of biasedness of the response. Using this more sophisticated evaluation metric may help to explain the reason why in some scenarios (as in Table 2) the vulnerability is lower under attacks compared to the No Attack case, which is an unexpected pattern. Some explanations on the choice of the evaluation metric will be helpful.

4. In Table 3, the results of cross-bias attacks based on Race-Religion and Gender-Religion pairs are not included, is there a particular reason for that? Also, from the results in Table 3, it seems that LLMs are more vulnerable to single-bias attacks, which is consistent with expectation. Given this observation, could you please clarify the potential significance (usefulness) of cross-bias attacks?

**Questions:**

Please see the weaknesses above.

I will raise my score if the authors could address my concerns.

---

### Official Review · Reviewer_Tnz4 · 2024-11-08

**Soundness:** 3
**Presentation:** 3
**Contribution:** 1
**Rating:** 3
**Confidence:** 3

**Summary:**

In this paper the authors study if jailbreak methods can be used to cause LLMs to give more socially/demographically biased outputs.  They test 3 different jailbraeking techniques against a large number of models and make a number of observations about which cases are more susceptible.

**Strengths:**

* The paper runs a very large and seemingly exhaustive set of experiments to do their analysis.
* There are intermediate contributions such as the creation of a dataset of prompts for biased output that I think could be useful (if released).

**Weaknesses:**

* The biggest question for me is the alignment of motivation and experiments -- the motivation for an adversary trying to trick a model to make a biased statement seems more synthetic.  It'd be interesting to push this into more realistic settings such as writing hateful content (eg tweets).

* The definitions of attack success seems quite loose and not able to differentiate between slightly biased and highly harmful content.  Many benchmarks and platforms now have more nuanced definitions of unacceptable content that would be valuable to anchor on (eg ML Commons's "hate" definition).

* Generally, while I appreciate the thoroughness of the work, the novelty is limited.

* While not core to the paper, the defense methods chosen (all prompt based) are fairly weak.

**Questions:**

* Can you share the prompts themselves? It is quite hard to tell how egregious they are.

* Likewise the human annotation of whether it is biased seems like it could range from really bad outputs to slightly biased. Is there more detail here?

---

### Note · Authors · 2024-11-26

I have read and agree with the venue's withdrawal policy on behalf of myself and my co-authors.